# UNLEARNING TRAINING DATA FROM DIFFUSION MODELS

## ABSTRACT

Diffusion models (DMs) have demonstrated remarkable generative capabilities in image generation but also pose privacy and copyright risks by memorizing and exposing training images. This concern is heightened by privacy regulations such as GDPR, which grant individuals the right to request the deletion of their data from AI models. Machine unlearning (MU) has been proposed to address this issue, as it enables the selective removal of specific training data from AI models. However, most existing MU methods for DMs primarily focus on unlearning at the class level—either by removing entire classes of data or class-specific features. In contrast, sample-level machine unlearning (SLMU), which targets the removal of individual training samples, remains an underexplored area. SISS is the pioneering work on SLMU for DMs. However, after careful investigation, we find that the evaluation metric used in SISS does not adequately assess unlearning performance. Moreover, under our proposed evaluation framework, SISS cannot achieve complete unlearning and presents significant degradation in generative performance. In this paper, we first define the objective of SLMU for DMs. Building on this definition, we introduce a quantitative evaluation framework for constructing benchmarks that compare different methods. Using this framework, we are the first to identify the *fake unlearning* phenomenon. Additionally, we propose a novel Sample-Level Machine Unlearning approach for Diffusion models, termed SMUD. SMUD alters the generative path of the targeted images, leading the DM to generate different images. Quantitative experimental results against baselines demonstrate that the proposed SMUD is the only method that can achieve SLMU without fake unlearning for both unconditional and conditional DMs.

## 1 INTRODUCTION

Diffusion models (DMs) have gained significant attention as powerful generative models. Trained on large-scale image datasets, DMs generate high-fidelity images that align closely with the training data distribution. DMs are broadly classified into two types: unconditional and conditional. Unconditional DMs generate high-quality images from Gaussian random inputs without additional information Ho et al. (2020); Nichol & Dhariwal (2021). In contrast, conditional DMs leverage auxiliary information to guide the generation process, enabling tasks such as text-to-image generation Rombach et al. (2022); Ramesh et al. (2022); Saharia et al. (2022b) and image-to-image translation Meng et al. (2022); Lugmayr et al. (2022); Saharia et al. (2022a).

However, alongside their impressive generative capabilities, DMs also raise privacy concerns; for instance, both unconditional and conditional DMs can generate duplicates of training images Wang et al. (2024b); Somepalli et al. (2023); Carlini et al. (2023); Chen et al. (2024b), leading to privacy breaches and copyright infringement. DMs present strong memorization, i.e., most generated data are duplicates of the training data when trained on small datasets Yoon et al. (2023); Baptista et al. (2025); Gu et al. (2023); Zhang et al. (2024b). Currently, some regulations, such as the General Data Protection Regulation (GDPR) GDP (2016) and the California Consumer Privacy Act (CCPA) CCP (2018), are addressing privacy and copyright risks. These regulations grant individuals the right to request the deletion of their data from a well-trained AI model.

Machine unlearning (MU) Bourtoule et al. (2021) has been proposed to remove the training data from a trained AI model to ensure privacy and copyright compliance. MU solutions can be catego-

rized into exact unlearning and approximate unlearning Thudi et al. (2022). Exact unlearning Bourtoule et al. (2021); Yan et al. (2022) removes the training data from the model through algorithmic-level retraining and applies ensemble models for acceleration. Approximate unlearning Mehta et al. (2022); Golatkar et al. (2020b); Neel et al. (2021) aims to minimize the influence of targeted data points to an acceptable level rather than completely removing them. However, the above-mentioned unlearning methods are designed for classification models and unsuitable for DMs. Details about MU for classification models are available in Appendix A.1. On the other hand, most existing MU methods for DMs are primarily focus on class-level MU, such as unlearning an entire class of data or a class of features Zhang et al. (2024c;a); Li et al. (2024a); Fan et al. (2024); Fuchi & Takagi (2024); Gandikota et al. (2023). These methods cannot solve the finer-grained sample-level machine unlearning (SLMU) since they require a conditioning input and unlearn all features related to the input. Details about MU for DMs are available in Appendix A.2.

Another challenge of SLMU for DMs lies in defining appropriate evaluation metrics that measure how well the DMs unlearn the targeted samples. For class-level MU in DMs, we can evaluate the performance of MU methods by verifying whether the unlearned DMs generate the targeted classes or features, but this evaluation is not suitable for SLMU. On the other hand, evaluation metrics for SLMU in classification models are usually based on the model's output Chundawat et al. (2023); Fan et al. (2024); Chen et al. (2023a); Foster et al. (2024); Kurmanji et al. (2024); Liu et al. (2024). However, these metrics are unsuitable for DMs, as the output of a DM is a random three-dimensional vector rather than a deterministic classification vector. Furthermore, the distributions of the generated images before and after unlearning may appear indistinguishable Stadler et al. (2022); Yuan et al. (2024a;b) since the targeted unlearning data are in-distribution for SLMU. SISS Alberti et al. (2025) is the pioneering work in SLMU for DMs. However, after careful investigation (detail in Section 3.2), we find that the evaluation metric used in SISS is inadequate. Besides, SISS is built on a problematic assumption, i.e., fine-tuning on $X \setminus A$ can unlearn the unlearning set $A$ from the pretrained model, as Baseline-F in Fig. 7 and 14 in Appendix cannot achieve complete unlearning. Under our proposed evaluation framework, SISS fails to achieve complete unlearning and exhibits significant degradation in generative performance. To address these research gaps, this paper introduces a novel method—Sample-level Machine Unlearning for Diffusion models, termed as SMUD. SMUD alters the generative path of the targeted unlearning images, causing the DM to generate different images from those initially presented. Besides, this paper proposes a quantitative evaluation framework based on the memorization property of DMs, which can be used to construct a benchmark and thus provides a foundation for future research.

In summary, the paper makes the following contributions:

- We propose a ***novel quantitative evaluation framework*** for SLMU in DMs, leveraging DMs' memorization property, which can be used to construct a benchmark and thus provides a foundation for future research.

- To the best of our knowledge, we are the *first* to observe the *fake unlearning* phenomenon in machine unlearning and incorporate it into the proposed evaluation framework.

- We propose ***a novel SLMU method***, SMUD, which intentionally changes the generation path of the targeted images to avoid generating them.

- We provide a comprehensive evaluation of the proposed SMUD. Quantitative results against four baselines demonstrate that our proposed SMUD is the only method to achieve SLMU without fake unlearning for both unconditional and conditional DMs.

## 2 PRELIMINARIES FOR DIFFUSION MODELS

The DMs introduced in this section are based on DDPM Ho et al. (2020). DDPM operates in two stages, i.e., the forward and reverse processes. The forward process starts from clean data $\mathbf{x}_0$ and iteratively adds Gaussian noise to the data for $T$ steps until the data $\mathbf{x}_T$ becomes nearly indistinguishable from pure Gaussian noise. Given a forward process step $t$, $\mathbf{x}_t$ can be calculated by,

$$\mathbf{x}_t(\mathbf{x}_0, \boldsymbol{\epsilon}) = \sqrt{\bar{\alpha}_t}\mathbf{x}_0 + \sqrt{1 - \bar{\alpha}_t}\boldsymbol{\epsilon} \text{ for } \boldsymbol{\epsilon} \sim \mathcal{N}(\mathbf{0}, \mathbf{I}), \tag{1}$$

where $\bar{\alpha}_t$ is a pre-defined parameter and $t \in \{1, 2, \cdots, T\}$. The reverse process starts from pure Gaussian noise $\hat{\mathbf{x}}_T \sim \mathcal{N}(\mathbf{0}, \mathbf{I})$ and iteratively denoises the data with the estimated noise, $\hat{\boldsymbol{\epsilon}} =$

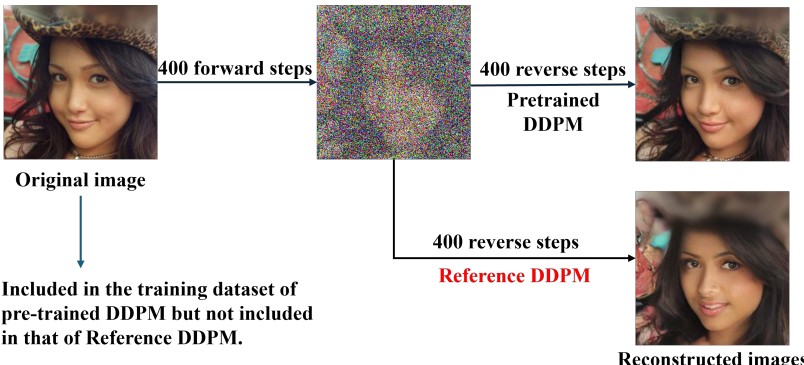

Figure 1: Demonstration of memorization of DMs where the original image is not included in the training dataset of reference DDPM but included in that of pre-trained DDPM.

$\epsilon_{\boldsymbol{\theta}}(\hat{\mathbf{x}}_t, t)$, where $\epsilon_{\boldsymbol{\theta}}$ is a trainable approximator, until getting $\hat{\mathbf{x}}_0$. Given $\hat{\mathbf{x}}_t$, $\hat{\mathbf{x}}_{t-1}$ can be calculated by,

$$\hat{\mathbf{x}}_{t-1} = \frac{1}{\sqrt{\alpha_t}}\left(\hat{\mathbf{x}}_t - \frac{1-\alpha_t}{\sqrt{1-\bar{\alpha}_t}}\epsilon_{\boldsymbol{\theta}}\left(\hat{\mathbf{x}}_t, t\right)\right) + \sigma_t \mathbf{z}_t, \tag{2}$$

where $t \in \{1, 2, \cdots, T\}$, $\mathbf{z}_t \sim \mathcal{N}(\mathbf{0}, \mathbf{I})$, and $\sigma_t$ and $\alpha_t$ are pre-defined parameters. To make the distribution of $\hat{\mathbf{x}}_0$ similar to that of $\mathbf{x}_0$, the deep learning-based approximator $\epsilon_{\boldsymbol{\theta}}$ is optimized according to the following loss function,

$$L(\epsilon_{\boldsymbol{\theta}}) = \mathbb{E}_{t, \mathbf{x}_0, \epsilon}\left[\left\|\epsilon - \epsilon_{\boldsymbol{\theta}}\left(\sqrt{\bar{\alpha}_t}\mathbf{x}_0 + \sqrt{1-\bar{\alpha}_t}\epsilon, t\right)\right\|^2\right]. \tag{3}$$

On the other hand, conditional DDPM incorporates conditioning input into the reverse process enabling more targeted and controlled generation. The forward process of conditional DDPM is the same as **Eq.**(1). But it considers the conditioning input $\mathbf{c}$ to estimate the noise at the step $t$ of the reverse process. Given $\hat{\mathbf{x}}_t$ in the reverse process of conditional DDPM, $\hat{\mathbf{x}}_{t-1}$ can be calculated by,

$$\hat{\epsilon} = \epsilon_{\boldsymbol{\theta}}\left(\hat{\mathbf{x}}_t, t, \emptyset\right) + \beta\left(\epsilon_{\boldsymbol{\theta}}\left(\hat{\mathbf{x}}_t, t, \mathbf{c}\right) - \epsilon_{\boldsymbol{\theta}}\left(\hat{\mathbf{x}}_t, t, \emptyset\right)\right),$$
$$\hat{\mathbf{x}}_{t-1} = \frac{1}{\sqrt{\alpha_t}}\left(\hat{\mathbf{x}}_t - \frac{1-\alpha_t}{\sqrt{1-\bar{\alpha}_t}}\hat{\epsilon}\right) + \sigma_t \mathbf{z}_t, \tag{4}$$

where $\mathbf{c}$ is the conditioning input, $\emptyset$ is the feature for the null condition that is usually a zero vector, and $\beta$ is a scalar for the conditional scale. Accordingly, the loss function of conditional DDPM is calculated by,

$$L_c(\epsilon_{\boldsymbol{\theta}}) = \mathbb{E}_{t, \mathbf{x}_0, \epsilon}\left[\left\|\epsilon - \epsilon_{\boldsymbol{\theta}}\left(\sqrt{\bar{\alpha}_t}\mathbf{x}_0 + \sqrt{1-\bar{\alpha}_t}\epsilon, t, \mathbf{c}\right)\right\|^2\right]. \tag{5}$$

## 3 EVALUATION FRAMEWORK

In this section, we first define the objectives of SLMU in DMs and then present the proposed evaluation framework based on this definition.

### 3.1 OBJECTIVE DEFINITION

The objective of SLMU for general machine learning models has been defined in Bourtoule et al. (2021) based on the distribution of model parameters. However, the distribution of model parameters is hard to measure for evaluation purposes. Moreover, we need to train multiple DMs to estimate the parameter distribution, which is unrealistic considering the computational requirements of training a DM. On the other hand, the provider of image generation service usually keeps the model parameters private and releases an API for users to generate synthetic images using a well-trained DM. In this scenario, potential privacy breaches and the impact of individual data are primarily manifested through the generated images. In this regard, Definition 1 defines the objective of SLMU for DMs based on the distribution of generated images.

**Definition 1** *Let $\boldsymbol{\theta}_p$ denote a pretrained DM that is trained on dataset $D$. Select a subset $D_u$ from $D$ as the unlearning set and retain set $D_r = D \setminus D_u$. Given an SLMU mechanism $\mathcal{M}$, the unlearned DM $\boldsymbol{\theta}_u$ is obtained by $\boldsymbol{\theta}_u = \mathcal{M}(\boldsymbol{\theta}_p, D_r, D_u)$. Let $P_u$ denote the distribution of the images generated by $\boldsymbol{\theta}_u$. The objectives of $\mathcal{M}$ are to (i) increase the similarity between $P_u$ and the distribution of the retain set and (ii) erase the memorization of $D_u$ in $\boldsymbol{\theta}_p$.*

**Note:** Same as Alberti et al. (2025), the size of the unlearning set $D_u$ is assumed to be much smaller than that of the training dataset $D$. As a result, $D_r$ has a similar distribution to $D$.

### 3.2 PROPOSED EVALUATION FRAMEWORK

In this Section, we first present challenges in evaluation metrics for SLMU and describe the problems with the evaluation metrics proposed in Alberti et al. (2025). Then, the proposed quantitative evaluation framework is introduced.

**Challenges in evaluation metrics.** As outlined in Definition 1, SLMU in DMs has two primary objectives. Objective (i) can be effectively evaluated using the FID Heusel et al. (2017). However, objective (ii) presents a significant challenge—there is no clear method to definitively determine if the memorization of $D_u$ has been fully unlearned. The main reason is that the distribution of the training dataset $D$ is indistinguishable from that of the retain set $D_r$ under practical metrics, as the size of the unlearning set $D_u$ is small and its samples are in-distribution. Therefore, if objective (i) is met, the distributions of generated data before and after unlearning are indistinguishable. This phenomenon has also been observed in Stadler et al. (2022); Yuan et al. (2024a;b).

**Problems with existing evaluation metrics.** In Fig. 1, the original image $\mathbf{x}_{ori}$ is part of the training dataset $D$ for the pretrained DDPM, while the reference DDPM is trained on $D \setminus \{\mathbf{x}_{ori}\}$. In this setup, the reference DDPM serves as the ideal unlearned model for the pretrained DDPM, where the unlearning set $D_u = \{\mathbf{x}_{ori}\}$. To evaluate the memorization, we generate a noised image with 400 forward steps, which retains partial information from the original image. **Note** that after 1000 forward steps, the image becomes pure Gaussian noise and is non-reconstructable. We then conduct 400 reverse steps to reconstruct the image using both the pretrained and reference DDPMs. As depicted in Fig. 1, the pretrained DDPM reconstructs an image that is nearly identical to the original, while the reference DDPM reconstructs an image that slightly differs from the original. The authors of SISS Alberti et al. (2025) argue that a larger difference between the original and reconstructed images indicates more effective unlearning. However, this definition is problematic. As shown in Fig. 1 and Fig. 3, although the reference DDPM has never seen the original image during training, it can reconstruct an image that closely resembles the original image. Since the reference model is the ideal unlearned model, we cannot conclude that a larger difference between the original and reconstructed images necessarily means more effective unlearning.

**Proposed quantitative evaluation metric.** We use the strong memorization of DMs for evaluation instead of the process demonstrated in Fig. 1 used in Alberti et al. (2025). First, a DM is pretrained on a small dataset $D$. Second, generate a synthetic dataset $\hat{D}_p$ with the pretrained DM. Third, select the replicates of $D$'s data from the synthetic dataset $\hat{D}_p$, using the same method as Yoon et al. (2023); Gu et al. (2023). The unlearning set $D_u$ is constructed by the $N$ most memorized training data. Then, we unlearn the pretrained DM and generate a synthetic dataset $\hat{D}_u$ with the unlearned DM. Last, we select (using the same method as Yoon et al. (2023); Gu et al. (2023)) and count the duplicates of $D_u$'s data from $\hat{D}_u$ as the quantitative evaluation metric, which is termed as Number of Duplicates of the Unlearning Set (NDUS). Smaller NDUS means better unlearning.

**Fake unlearning.** In our preliminary experiments, we observed that initially unlearned DMs, which do not generate duplicates of images from the unlearning set, start to generate such duplicates after fine-tuning on the retain set. This phenomenon is termed as fake unlearning. The unlearned DM initially avoids generating duplicates due to the performance degradation caused by the unlearning process, and fine-tuning the unlearned model on the retain set can recover the generative performance. Fake unlearning indicates that the DM does not completely forget the unlearning data. We incorporate this fake unlearning in our proposed evaluation framework by measuring NDUS after fine-tuning the unlearned DM on the retain set. The overall evaluation framework is summarized in Appendix B.

**Rationale behind NDUS.** According to Theorem 4.3 in Baptista et al. (2025), during the reverse process of DMs, if any intermediate result enters the Voronoi cell of a training sample, the generation trajectory will converge to that sample. Experimental results in Baptista et al. (2025) also show that DMs always generate duplicates of training data when the model has enough parameters. This finding is consistent with other works Yoon et al. (2023); Gu et al. (2023); Zhang et al. (2024b). Based on this theorem, if a DM stops generating previously memorized data, we can infer that it has unlearned that training sample. However, as discussed earlier, fake unlearning can occur due to a decline in generative performance during the unlearning process. To address this, we fine-tune the unlearned DM on the retain set to recover its generative capabilities and then check whether it generates the previously memorized data.

# 4 SAMPLE-LEVEL MACHINE UNLEARNING FOR DIFFUSION MODELS

In this section, we first provide a detailed introduction to SMUD for unconditional DMs and then briefly describe SMUD for conditional DMs, which is largely identical to the unconditional case.

## 4.1 SMUD FOR UNCONDITIONAL DMS

To achieve the objectives of SLMU, we first define the **noised reverse process** for unconditional DMs as follows,

$$\hat{\mathbf{x}}_{t-1} = \frac{1}{\sqrt{\alpha_t}} \left( \hat{\mathbf{x}}_t - \frac{1 - \alpha_t}{\sqrt{1 - \bar{\alpha}_t}} \left( \boldsymbol{\epsilon}_{\boldsymbol{\theta}}(\hat{\mathbf{x}}_t, t) + \gamma \boldsymbol{\epsilon}' \right) \right) + \sigma_t \mathbf{z}_t, \tag{6}$$

where $\gamma \in (0, \infty)$ is a coefficient controlling the noise amplitude, $\boldsymbol{\epsilon}' \sim \mathcal{N}(\mathbf{0}, \mathbf{I})$, and other parameters and variables are the same as **Eq.**(2). **Eq.**(6) can be rewritten as,

$$\hat{\mathbf{x}}_{t-1} = \frac{1}{\sqrt{\alpha_t}} \left( \hat{\mathbf{x}}_t - \frac{1 - \alpha_t}{\sqrt{1 - \bar{\alpha}_t}} \left( \boldsymbol{\epsilon}_{\boldsymbol{\theta}}(\hat{\mathbf{x}}_t, t) \right) \right) + \frac{\gamma(1 - \alpha_t)}{\sqrt{\alpha_t(1 - \bar{\alpha}_t)}} \boldsymbol{\epsilon}' + \sigma_t \mathbf{z}_t. \tag{7}$$

The summary of the last two terms in **Eq.**(7), i.e., $\frac{\gamma(1-\alpha_t)}{\sqrt{\alpha_t(1-\bar{\alpha}_t)}} \boldsymbol{\epsilon}' + \sigma_t \mathbf{z}_t$, follows a Gaussian distribution with mean value of $\mathbf{0}$ since both $\boldsymbol{\epsilon}'$ and $\mathbf{z}_t$ follows $\mathcal{N}(\mathbf{0}, \mathbf{I})$. Therefore, the noised reverse process **Eq.**(7) can be seen a standard reverse process **Eq.**(2) with a larger $\sigma_t$. According to the analysis in Kynkäänniemi et al. (2019), if $\gamma$ is properly chosen (neither too large nor too small), and given the same $\hat{\mathbf{x}}_T$, well-trained $\boldsymbol{\epsilon}_{\boldsymbol{\theta}}$, and $\mathbf{z}_t$, the noised and standard reverse processes will produce different images. This has been experimentally validated in Section 5.1.

To achieve SLMU, we use the noised reversed process to fine-tune the pretrained DM on the unlearning set. Specifically, we add a Gaussian noise to $\boldsymbol{\epsilon}_{\boldsymbol{\theta}}(\mathbf{x}_t, t)$ and use the result as the label to optimize $\boldsymbol{\epsilon}_{\boldsymbol{\theta}}(\mathbf{x}_t, t)$ when the input $\mathbf{x}_0$ is sampled from unlearning set $D_u$. The unlearning loss $L_u$ for unconditional DMs is calculated as,

$$L_u(\boldsymbol{\epsilon}_{\boldsymbol{\theta}}) = \mathbb{E}_{t, \mathbf{x}_0 \in D_u, \boldsymbol{\epsilon} \text{ and } \boldsymbol{\epsilon}' \sim \mathcal{N}(\mathbf{0}, \mathbf{I})}[\|\boldsymbol{\epsilon}'_{\boldsymbol{\theta}}(\mathbf{x}_t, t) + \gamma \boldsymbol{\epsilon}' - \boldsymbol{\epsilon}_{\boldsymbol{\theta}}(\mathbf{x}_t, t)\|^2], \tag{8}$$

where $\boldsymbol{\epsilon}'_{\boldsymbol{\theta}}$ is a copy of $\boldsymbol{\epsilon}_{\boldsymbol{\theta}}$ and not optimized during unlearning, $\gamma \in (0, \infty)$ controls the noise amplitude, $\boldsymbol{\epsilon}' \sim \mathcal{N}(\mathbf{0}, \mathbf{I})$, and other parameters are the same as **Eq.**(3). Optimizing $\boldsymbol{\epsilon}_{\boldsymbol{\theta}}$ by minimizing **Eq.**(8) alters the generation path of the unlearning set images to other images and thus achieve SLMU. Moreover, this unlearning loss does not significantly affect the generation performance, as the altered images remain within the distribution of the training dataset.

On the other hand, the distribution of the images generated by the unlearned model is required to be similar to the distribution of the retain set $D_r$ as discussed in Definition 1. To achieve this objective, when the input $\mathbf{x}_0$ is sampled from the retain set, we apply the original loss functions of unconditional DMs as the retain loss $L_r$, i.e.,

$$L_r(\boldsymbol{\epsilon}_{\boldsymbol{\theta}}) = \mathbb{E}_{t, \mathbf{x}_0 \in D_r, \boldsymbol{\epsilon} \sim \mathcal{N}(\mathbf{0}, \mathbf{I})}[\|\boldsymbol{\epsilon} - \boldsymbol{\epsilon}_{\boldsymbol{\theta}}(\sqrt{\bar{\alpha}_t}\mathbf{x}_0 + \sqrt{1 - \bar{\alpha}_t}\boldsymbol{\epsilon}, t)\|^2]. \tag{9}$$

We optimize the approximator by minimizing the unlearning loss **Eq.**(8) every $N_{\text{interval}}$ optimization steps to facilitate unlearning while by minimizing the retain loss **Eq.**(9) at each optimization step to preserve the model's generative capability.

## 4.2 SMUD FOR CONDITIONAL DMS

Similar to the unconditional DDPM, the noised reverse process for conditional DMs is defined as,

$$\hat{\epsilon} = \epsilon_{\boldsymbol{\theta}}\left(\hat{\mathbf{x}}_t, t, \emptyset\right) + \beta\left(\epsilon_{\boldsymbol{\theta}}\left(\hat{\mathbf{x}}_t, t, \mathbf{c}\right) - \epsilon_{\boldsymbol{\theta}}\left(\hat{\mathbf{x}}_t, t, \emptyset\right)\right),$$

$$\hat{\mathbf{x}}_{t-1} = \frac{1}{\sqrt{\alpha_t}}\left(\hat{\mathbf{x}}_t - \frac{1-\alpha_t}{\sqrt{1-\bar{\alpha}_t}}(\hat{\epsilon} + \gamma\epsilon')\right) + \sigma_t \mathbf{z}_t, \tag{10}$$

where $\gamma \in (0, \infty)$ is a coefficient controlling the noise amplitude and $\epsilon' \sim \mathcal{N}(\mathbf{0}, \mathbf{I})$. Similar to the unconditional case, the noised reverse process **Eq.**(10) can be seen a standard reverse process **Eq.**(4) with a larger $\sigma_t$. According to the analysis in Kynkäänniemi et al. (2019), if $\gamma$ is properly chosen (neither too large nor too small), and given the same $\hat{\mathbf{x}}_T$, well-trained $\epsilon_{\boldsymbol{\theta}}$, and $\mathbf{z}_t$, the noised and standard reverse processes will produce different images that follow a distribution similar to the training dataset, which has been validated experimentally in Section 5.1.

Unlike unconditional DMs, unlearning set images' information is encoded in $\epsilon_{\boldsymbol{\theta}}$ when the conditions are either $\mathbf{c}$ (the ground-truth condition) or $\emptyset$ (the null condition). To simulate the reverse process **Eq.**(10), the unlearning loss, $L_u$, for conditional DMs is calculated by,

$$\mathbf{y_c} = (\sqrt{\bar{\alpha}_t}\mathbf{x}_0 + \sqrt{1-\bar{\alpha}_t}\epsilon, t, \mathbf{c}), \quad \mathbf{y}_\emptyset = (\sqrt{\bar{\alpha}_t}\mathbf{x}_0 + \sqrt{1-\bar{\alpha}_t}\epsilon, t, \emptyset),$$

$$L_u(\epsilon_{\boldsymbol{\theta}}) = \mathbb{E}_{t,\mathbf{x}_0 \in D_u, \epsilon \text{ and } \epsilon' \sim \mathcal{N}(\mathbf{0}, \mathbf{I})}[\|\epsilon'_{\boldsymbol{\theta}}(\mathbf{y_c}) + \gamma\epsilon' - \epsilon_{\boldsymbol{\theta}}(\mathbf{y_c})\|^2 + \|\epsilon'_{\boldsymbol{\theta}}(\mathbf{y}_\emptyset) + \gamma\epsilon' - \epsilon_{\boldsymbol{\theta}}(\mathbf{y}_\emptyset)\|^2]. \tag{11}$$

Similar to the unconditional case, we need to optimize $\epsilon_{\boldsymbol{\theta}}$ according to the retain loss to maintain the generative capability of the DM. The retain loss, $L_r$, for conditional DMs is calculated as,

$$L_r(\epsilon_{\boldsymbol{\theta}}) = \mathbb{E}_{t,\mathbf{x}_0 \in D_r, \epsilon \sim \mathcal{N}(\mathbf{0}, \mathbf{I})}[\|\epsilon - \epsilon_{\boldsymbol{\theta}}(\sqrt{\bar{\alpha}_t}\mathbf{x}_0 + \sqrt{1-\bar{\alpha}_t}\epsilon, t, \mathbf{c})\|^2]. \tag{12}$$

The pipeline of SMUD is summarized in Appendix C.

## 5 EXPERIMENTS

In this section, we quantitatively evaluate SMUD with the proposed evaluation framework. Then, we qualitatively evaluate SMUD trained on a large dataset as demonstrated in Fig. 1.

### 5.1 EVALUATION OF NOISED REVERSE PROCESSES

To evaluate the effectiveness of the noised reverse process **Eq.**(6) for unconditional DMs, we apply the pretrained $\boldsymbol{\theta}_p$ to generate images using standard reverse process, i.e., $\gamma = 0$ in **Eq.**(6), and noised reverse process, i.e., $\gamma > 0$ in **Eq.**(6). As shown in the left column of Fig. 2, when $\gamma$ is small, e.g., $\gamma = 0.05$, the images generated by the standard and noised reverse processes are almost the same. When $\gamma$ becomes larger, e.g., $\gamma = 0.1$, the images generated by the standard and noised reverse processes become more different. Note that $\mathbf{z}_t$ and $\hat{\mathbf{X}}_T$ in **Eq.**(2) keep the same by using the same random seed across different $\gamma$ values. These results demonstrate that injecting Gaussian noise into $\epsilon_{\boldsymbol{\theta}}(\hat{\mathbf{x}}_t, t)$ can alter the generated samples while ensuring that they remain consistent with the training distribution. Consequently, the proposed SMUD preserves the generative capability of the model after unlearning. Similar observations are obtained for conditional DMs, as illustrated in the right column of Fig. 2.

### 5.2 BASELINES AND EVALUATION METRICS.

In this paper, we consider four baselines. The first baseline is training the DM on $D_r$ from scratch, referred to as Baseline-R. In the field of continual learning Wang et al. (2024a), it has been observed that fine-tuning a pre-trained deep learning model on new data can degrade its performance on previously learned data. Building on this property, the second baseline involves fine-tuning the pre-trained DM on the retain dataset $D_r$, termed as Baseline-F. The third baseline is based on gradient ascent Huang et al. (2024), termed as Baseline-GA. Although the method in Huang et al. (2024) is initially proposed for class-level MU for DMs, it can be adapted to SLMU. The fourth baseline is SISS Alberti et al. (2025), which is the pioneering work to address SLMU in DMs.

For evaluation, we first use the unlearned DM to generate a synthetic dataset $\hat{D}_u$ of the same size as the pre-training dataset. We then compute the FID (denoted as FID_M) between the synthetic dataset

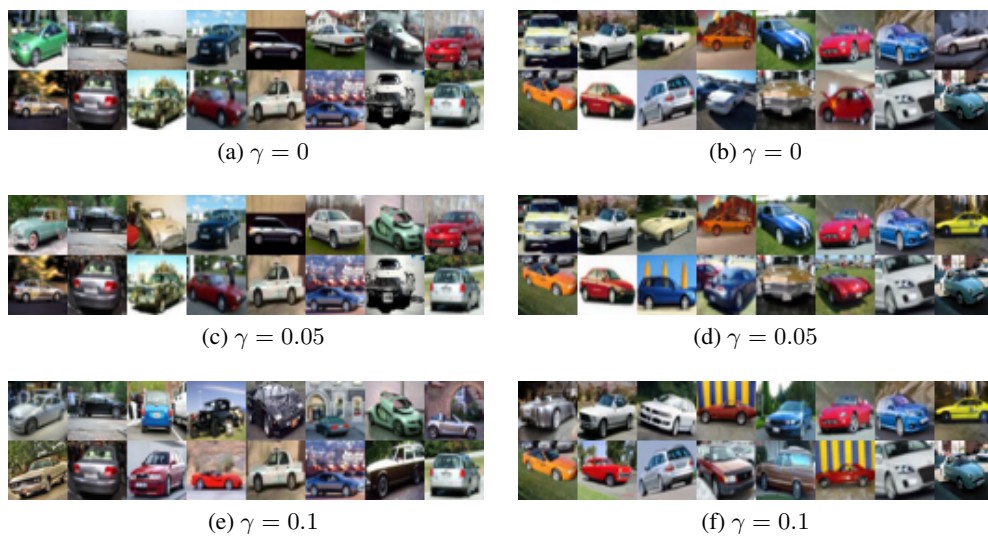

(a) $\gamma = 0$             (b) $\gamma = 0$

(c) $\gamma = 0.05$            (d) $\gamma = 0.05$

(e) $\gamma = 0.1$             (f) $\gamma = 0.1$

Figure 2: Images generated by the pretrained $\boldsymbol{\theta}_p$ with different $\gamma$ values in the noised reverse process **Eq.**(6) (left column) and **Eq.**(10) (right column).

$\hat{D}_u$ and retain set $D_r$ to evaluate objective (i) of Definition 1. Comparing the synthetic dataset $\hat{D}_u$ and unlearning set $D_u$, we select duplicates of $D_u$'s data from $\hat{D}_u$ using the method in Yoon et al. (2023); Gu et al. (2023) and count the duplicates as NDUS (denoted as NDUS_M) to evaluate the objective (ii) of Definition 1. Finally, to detect fake unlearning, we fine-tune the unlearned DM on $D_r$ and compute FID and NDUS again (denoted as FID_F and NDUS_F, respectively).

## 5.3 EXPERIMENTS FOR UNCONDITIONAL DMS

We apply the unconditional DDPM proposed in Nichol & Dhariwal (2021) with a linear noise schedule and 1K diffusion steps. We randomly select 2K images from each of the five classes in the CIFAR-10 dataset—automobile, airplane, bird, cat, and deer—as five separate training sets. We pre-train the DDPM for 1M steps with the mini-batch size 128 on a training set and obtain the pre-trained DDPM $\boldsymbol{\theta}_p$, and then use it to generate a synthetic dataset $\hat{D}_p$ with 2K images. Then we select 16 images in $D$, which have the most duplicates in $\hat{D}_p$, to construct the unlearning set $D_u$ and the remaining 1984 images of $D$ construct the retain set $D_r$. We set $N_{\text{interval}} = 5$ and $\gamma = 1.0$.

Table 1 presents a performance overview of the proposed SMUD and two baselines on unlearning five CIFAR-10 classes separately. For SMUD and Baseline-GA, the DMs are unlearned for 4K steps and subsequently finetuned for 4K steps to assess fake unlearning. For SISS, fewer unlearning steps are applied, as excessive steps severely degrade generation quality (see Appendix D); the unlearned model is then fine-tuned for 4K steps. As shown in Table 1, SMUD achieves the best generative performance after unlearning without exhibiting fake unlearning. In contrast, both Baseline-GA and SISS significantly reduce the generative quality and display fake unlearning. Finetuning the DMs unlearned by Baseline-GA and SISS results in a lower FID and higher NDUS, indicating that the unlearning ability of Baseline-GA and SISS is partly due to the decline in generative performance. Appendix E shows more detailed results during unlearning Automobile including Baseine-F.

## 5.4 EXPERIMENTS FOR CONDITIONAL DMS

We apply the DDPM with classifier-free guidance Ho & Salimans (2021) with a linear noise schedule and 1K diffusion steps. We randomly selected 1K images from each class of the CIFAR-10 dataset to construct $D$ for the evaluation framework. We pretrain the model for 1M steps with the mini-batch size 128 on $D$ and obtain the pre-trained DDPM $\boldsymbol{\theta}_p$, and then use it to generate 1K images of one class to construct $\hat{D}_p$. Then we select 16 images in $D$, which have the most duplicates in $\hat{D}_p$, to construct the unlearning set $D_u$ and the remaining 9984 images of $D$ construct the retain set $D_r$. We set $N_{\text{interval}} = 5$ and $\gamma = 0.1$.

Table 1: Performance Overview of the Proposed SMUD and Baselines on Unconditional DDPM

| Unlearning Object | Method | FID_M ($\downarrow$) | NDUS_M ($\downarrow$) | FID_F ($\downarrow$) | NDUS_F ($\downarrow$) |
|---|---|---|---|---|---|
| **Airplane** | Baseline-GA | 25.11 | 0 | 11.74 | 0 |
| | SISS | 91.43 | 12 | 11.36 | 22 |
| | SMUD (ours) | 12.42 | 0 | 11.37 | 0 |
| **Automobile** | Baseline-GA | 27.68 | 1 | 9.03 | 8 |
| | SISS | 86.78 | 3 | 10.39 | 25 |
| | SMUD (ours) | 9.02 | 0 | 8.61 | 0 |
| **Bird** | Baseline-GA | 22.30 | 2 | 13.17 | 1 |
| | SISS | 110.00 | 0 | 14.03 | 22 |
| | SMUD (ours) | 15.96 | 0 | 14.42 | 0 |
| **Cat** | Baseline-GA | 36.97 | 0 | 16.23 | 0 |
| | SISS | 155.99 | 0 | 16.04 | 36 |
| | SMUD (ours) | 16.14 | 0 | 15.71 | 0 |
| **Deer** | Baseline-GA | 26.32 | 4 | 11.18 | 4 |
| | SISS | 87.88 | 0 | 10.80 | 25 |
| | SMUD (ours) | 12.19 | 0 | 10.99 | 0 |

Table 2: Performance Overview of the Proposed SMUD and Baselines on Conditional DDPM

| Unlearning Object | Method | FID_M($\downarrow$) | NDUS_M($\downarrow$) | FID_F($\downarrow$) | NDUS_F($\downarrow$) |
|---|---|---|---|---|---|
| **Airplane** | Baseline-GA | 57.23 | 0 | 57.48 | 1 |
| | SISS | 134.06 | 0 | 56.02 | 40 |
| | SMUD (Ours) | 63.85 | 0 | 55.40 | 0 |
| **Automobile** | Baseline-GA | 39.00 | 0 | 28.69 | 20 |
| | SISS | 112.13 | 0 | 30.30 | 35 |
| | SMUD (Ours) | 45.80 | 0 | 29.10 | 0 |
| **Bird** | Baseline-GA | 80.85 | 1 | 44.58 | 15 |
| | SISS | 146.27 | 1 | 44.24 | 45 |
| | SMUD (Ours) | 54.56 | 0 | 42.41 | 0 |
| **Cat** | Baseline-GA | 69.25 | 0 | 54.56 | 53 |
| | SISS | 150.65 | 0 | 54.29 | 50 |
| | SMUD (Ours) | 63.03 | 0 | 51.99 | 0 |
| **Deer** | Baseline-GA | 54.34 | 0 | 49.75 | 10 |
| | SISS | 98.09 | 0 | 50.04 | 67 |
| | SMUD (Ours) | 60.85 | 0 | 46.49 | 0 |

Table 2 provides a performance overview of the proposed SMUD and two baselines across five CIFAR-10 classes. We apply SMUD and and Baseline-GA to unlearn the Automobile for 10K steps and each of the other four classes for 20K steps. The number of finetuning steps is set equal to the corresponding unlearning steps. For SISS, the unlearning is performed for approximately 50 steps, followed by 10K finetuning steps. As shown in Table 1, SISS leads to a substantial degradation in generative performance, while both SISS and Baseline-GA exhibit pronounced fake unlearning. SMUD maintains the best generative performance after unlearning and does not exhibit fake unlearning. Appendix F shows more detailed results during unlearning Automobile including Baseine-F.

## 5.5 EXPERIMENTS ON CELEBA-HQ DATASET

We use the CelebA-HQ dataset Odhiambo (2024), which contains 30K $256 \times 256$ images, to assess the proposed SMUD when the DM is trained on large datasets. We pre-train the unconditional DDPM on CelebA-HQ for 800K steps with the mini-batch size 32. We randomly select 128 images from CelebA-HQ to construct the unlearning set. The remaining 29,872 images construct the retain set. In this section, we present a qualitative evaluation as demonstrated in Fig. 1. We set $N_{\text{interval}} = 10$ and $\gamma = 1.0$. We find that the generative performance is significantly damaged after only 90 unlearning steps of SISS, as shown in Appendix D. Thus, we only qualitatively evaluate Baseline-R, Baseline-GA, and SMUD. The optimization steps for Baseline-GA and SMUD are 100K.

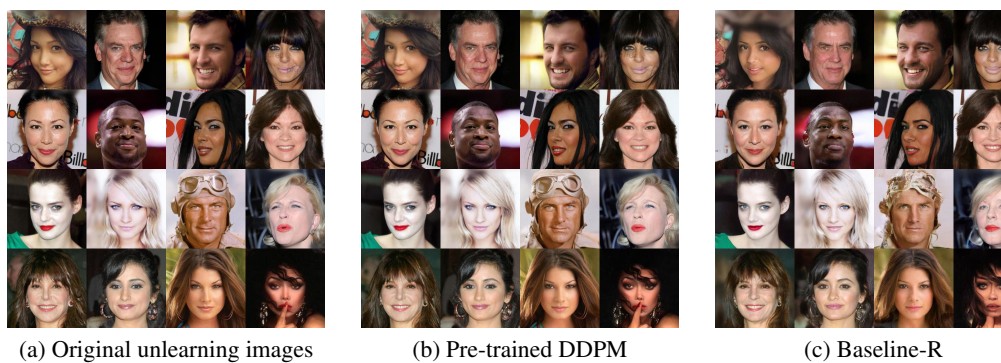

| (a) Original unlearning images | (b) Pre-trained DDPM | (c) Baseline-R |
|---|---|---|

Figure 3: 16 randomly selected unlearning set images and corresponding reconstructed images by the pre-trained DDPM and Baseline-R.

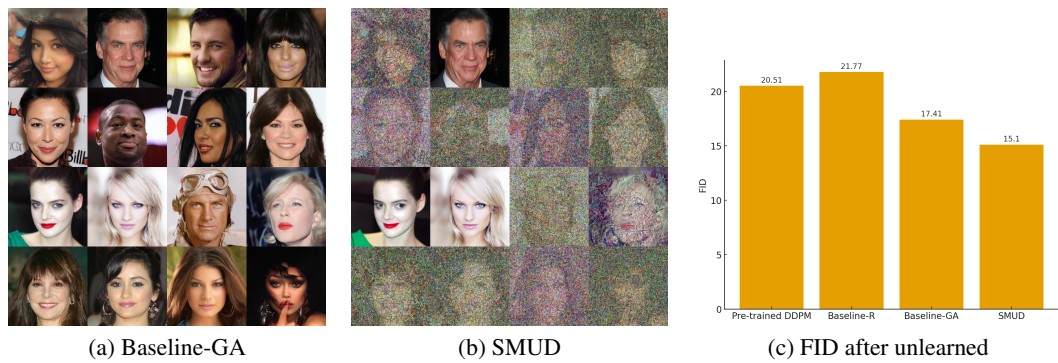

| (a) Baseline-GA | (b) SMUD | (c) FID after unlearned |
|---|---|---|

Figure 4: Reconstructed images by Baseline-GA and SMUD, and FID of DMs after being unlearned.

Figure 3a presents 16 images randomly selected from the unlearning set. Figures 3b and 3c show the corresponding reconstructions by the pre-trained DDPM and Baseline-R, respectively, where Baseline-R serves as the ideal unlearned DM. As illustrated in Fig. 3, the reconstructions generated by the pre-trained DDPM are nearly identical to the original images, while those from Baseline-R also exhibit high similarity. Figure 4 shows the reconstructed images generated by the DMs unlearned with Baseline-GA and SMUD. Comparing Figs. 4 and 3, the reconstructions from Baseline-GA visually resemble the original images more closely than those from Baseline-R. In contrast, the reconstructions produced by SMUD exhibit greater deviations, underscoring its effectiveness over Baseline-GA. Qualitative results of all 128 unlearning set images can be found in Appendix G. On the other hand, SMUD preserves the best generative performance as shown in Fig. 4c, suggesting that its superior unlearning performance is not due to any degradation in generative quality. Moreover, the DM exhibits higher generative performance after unlearning with SMUD compared to the pre-trained DM, indicating that SMUD minimally impacts generative quality.

## 6 CONCLUSION

In this paper, we first define two objectives for sample-level machine unlearning (SLMU) in diffusion models (DMs). We then propose a quantitative evaluation framework that leverages the memorization property of DMs to assess these objectives. This framework can be used to construct a benchmark for SLMU and thus lay a foundation for future research. Compared to the evaluation metrics proposed in the pioneering work on SLMU, our proposed evaluation framework provides a better assessment of unlearning methods and validates whether these methods can achieve complete unlearning. Additionally, we propose Sample-level Machine Unlearning for Diffusion models (SMUD), which modifies the generation path of DMs to prevent the generation of images in the unlearning set. Experimental results against baselines show that SMUD is the only method that does not exhibit fake unlearning in both unconditional and conditional DMs. Furthermore, SMUD preserves the highest generative performance after unlearning.

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

## A  RELATED WORK

In this section, we first review existing SLMU methods for classification tasks and discuss why most are inadequate for SLMU in DMs. Next, we review existing class-level MU methods for DMs. Last, we present the current evaluation frameworks for class-level MU methods in DMs, concluding that they are unsuitable for evaluating SLMU methods.

## A.1 SAMPLE-LEVEL MACHINE UNLEARNING FOR CLASSIFICATION MODELS

**Exact unlearning methods:** Existing SLMU methods in classification models can be categorized into exact unlearning and approximate unlearning. Exact unlearning methods unlearn the specific data by retraining the model. In Bourtoule et al. (2021), the authors introduce the Sharding, Isolation, Slicing, and Aggregation (SISA) framework, a general approach for exact unlearning. SISA enables selective data removal by sharding, isolating, slicing, and aggregating training data, avoiding full model retraining. Building on this idea, Bourtoule et al. (2021); Yan et al. (2022); Chen et al. (2022a) utilize ensemble methods that split the dataset into smaller sub-datasets, training a separate model for each. The final classification is based on the combined outputs of these models. To lower computational costs, they delete the data to be unlearned from the relevant sub-datasets and retrain only the affected sub-models. Schelter et al. (2021); Brophy & Lowd (2021) apply the SISA framework to tree-based classification models, while Chen et al. (2022b) extends SISA for Graph Neural Networks (GNNs). However, the SISA framework is specifically designed for classification models that work with partitionable datasets, allowing the unlearning set to be isolated. In contrast, generative models operate by learning distributions rather than mapping inputs to outputs directly, as in classification tasks. The SISA framework, focusing on classification models, is less suitable for generative models since less data will lead to significant performance decreases, and the ensemble model in SISA cannot improve the generation capability compared to one DM trained on one small dataset.

**Approximate unlearning methods:** Approximate unlearning reduces the computational cost of exact unlearning and includes methods based on (1) influence function estimation, (2) model parameter re-optimization, and (3) gradient updates Xu et al. (2024). Influence function, introduced by Guo et al. (2020), estimates the influence of given training data using the first-order gradient and second-order gradient (Hessian matrix) according to the loss function of those data and is used to remove the data's influence on the model. Besides, random noise is added to the parameters's gradients during optimization to remove the influence completely. However, this method relies on convexity assumptions and involves costly Hessian matrix inversion, and random noise decreases the model's performance. Later research Sekhari et al. (2021); Suriyakumar & Wilson (2022); Mehta et al. (2022); Wu et al. (2022) developed more efficient approximations. Model re-optimization methods, like weight perturbation, partially retrain models to update parameters without full retraining Golatkar et al. (2020a;b; 2021), though they still involve Hessian approximations. However, in DMs, inputs combine images with random Gaussian noise, and the model learns to predict this noise. Since the noise varies with each training step, calculating the Hessian for a specific image and noise is ineffective for unlearning. The noise used for Hessian estimation differs from that used during training, making the Hessian irrelevant for capturing the influence of a particular training sample. Gradient-based unlearning methods generally follow two steps: (1) initialize model parameters from the previously trained model, and (2) apply a few gradient updates based on modified data. DeltaGrad Wu et al. (2020) adapts models efficiently to small training dataset changes by using cached gradients and parameter information. However, it is impractical due to the large memory required to store this information for every training iteration.

## A.2 MACHINE UNLEARNING FOR DIFFUSION MODEL

DMs often utilize diverse open-source data, which can lead to the risk of incorporating sensitive or inappropriate information Chen et al. (2023b). This has raised concerns about the potential for generating harmful content Schramowski et al. (2023); Rando et al. (2022), violating copyright through the imitation of artistic styles Gandikota et al. (2023); Salman et al. (2023), or even memorizing training data Wang et al. (2024b); Somepalli et al. (2023); Carlini et al. (2023). Recent machine unlearning efforts in DMs focus on removing specific features or classes. For instance, Salun Fan et al. (2024) enables unlearning by fine-tuning only the most affected salient weights. Random labelling of the data from the unlearning set is used to update the salient weights. Forget-Me-Not Zhang et al. (2024a) leverages cross-attention scores to optimize the model's perception of target concepts. By optimizing vision-only self-attentive layers of stable diffusion using <nude, mosaic, benign> image triplets, Safegen Li et al. (2024b) remove pornographic latent representations from its attentive matrices, cutting off the associations between sexually-related text and nudity vision. In contrast, Fuchi & Takagi (2024) focuses on unlearning target concepts from the text encoder of Stable Diffusion via a gradient-ascent method without modifying the U-Net parameters. Erased Stable Diffusion

(ESD)Gandikota et al. (2023) fine-tunes model to align the conditional scores of undesired concepts with those of unconditioned, permanently removing learned concepts. Follows similar idea Chen et al. (2024a) aligns conditional scores of undesirable classes with those of safe classes. Wu et al. (2024b) frames the unlearning task as an adversarial training process, where the DM serves as the generator to predict noise, and a discriminator classifies whether the noise is linked to the target concept or the anchor concept. The objective is to align the DM's output between these two concepts. In Li et al. (2024a) and Wu et al. (2024a), the authors modify the reverse process of DMs by updating the loss function to align the predicted noise of specific concepts with a predefined noise distribution. However, the methods mentioned above are designed for conditional DMs to unlearn a class of data or feature and cannot solve the finer-grained sample-level machine unlearning. Sfront Huang et al. (2024) utilizes gradient ascent to achieve class-level machine unlearning. Although the method in Huang et al. (2024) is initially proposed for class-level MU, it can be adapted to SLMU. In this paper, we employ Sfront as a baseline, termed as Baseline-GA, since it can be adapted to solve SLMU in DMs.

## B    EVALUATION FRAMEWORK

**Overall evaluation framework.** Algorithm 1 summarizes the overall process of the proposed evaluation framework. (i) Train a DM $\theta_p$ on a dataset $D$ as the pre-trained model. (ii) Generate a synthetic dataset $\hat{D}_p$ (the same size as $D$) with $\theta_p$. (iii) Find memorized training images from $\hat{D}_p$ using the same method as Yoon et al. (2023); Gu et al. (2023). (iv) Select $N_u$ most memorized training images to construct the unlearning set $D_u$ and the retain set $D_r = D \setminus D_u$. (v) Apply an unlearning method $\mathcal{M}$ to obtain an unlearned DM $\theta_u = \mathcal{M}(\theta_p, D_r, D_u)$. (vi) Construct a synthetic dataset $\hat{D}_u$ (the same size as $D$) with $\theta_u$. (vii) Select duplicates of $D_u$'s data from $\hat{D}_u$ using the method in Yoon et al. (2023); Gu et al. (2023) and count the duplicates as NDUS. (viii) Fine-tune the unlearned DM $\theta_u$ on the retain set and calculating NDUS of the fine-tuned DM, similar to steps (vi) and (vii). (ix) Last, the proposed evaluation framework returns NDUS after unlearning, NDUS after finetuning, FID between $D_r$ and $\hat{D}_u$. The FID between $\hat{D}_u$ and $D_r$ evaluates objective (i) of Definition 1. NDUSs of unlearned DM and fine-tuned DM evaluate objective (ii) of Definition 1.

---

**Algorithm 1:** Evaluation framework.

**Input** : Dataset $D$; initialized DM $\theta_0$; training algorithm of DM $\mathcal{T}$; MU method $\mathcal{M}$.
**Output:** NDUS and FIDs

1 $\theta_p \leftarrow \mathcal{T}(\theta_0, D)$ `// Train `$\theta_p$` on D according to `**`Eq.(3)`**;
2 $\hat{D}_p \leftarrow \mathrm{DM}(\theta_p, \mathbf{z_1}), \mathbf{z_1} \in \mathcal{N}(\mathbf{0}, \mathbf{I})$ `// Generated `$\hat{D}_p$` with `$\theta_p$;
3 Select duplicates of $D$'s data from $\hat{D}_p$ using the method in Yoon et al. (2023); Gu et al. (2023)
4 Construct unlearning set $D_u$ with $N_u$ most memorized data
5 $D_r = D \setminus D_u$ `// Construct retain set`;
6 $\theta_u = \mathcal{M}(\theta_p, D_u, D_r)$ `// Obtain unlearned model `$\theta_u$;
7 $\hat{D}_u \leftarrow \mathrm{DM}(\theta_u, \mathbf{z_2}), \mathbf{z_2} \in \mathcal{N}$ `// Generated `$\hat{D}_u$` with `$\theta_u$;
8 Select duplicates of $D_u$'s data from $\hat{D}_u$ using the method in Yoon et al. (2023); Gu et al. (2023) and count the duplicates as NDUS
9 $\theta_f = \mathcal{T}(\theta_u, D_r)$ `// Finetune `$\theta_u$` on `$D_r$;
10 $\hat{D}_f \leftarrow \mathrm{DM}(\theta_u, \mathbf{z_3}), \mathbf{z_3} \in \mathcal{N}$ `// Generated `$\hat{D}_f$` with `$\theta_f$;
11 Compute NDUS by comparing $\hat{D}_f$ and $D_u$
12 **Return** NDUS after unlearning, NDUS after finetuning, $\mathrm{FID}(D_r, \hat{D}_u)$

---

## C    PSEUDO CODE OF SMUD

Algorithm 2 summarizes the proposed SMUD: (i) Sample images from the retain set and calculate the retain loss **Eq.**(9). (ii) If the optimization step $n$ satisfies $n\%N_{\mathrm{interval}} = 0$, copy the pre-trained model $\epsilon_\theta$ into $\epsilon'_\theta$ and sample images from the unlearning set. (iii) Calculate the unlearning loss **Eq.**(8) and add it to the retain loss. (iv) Optimize $\epsilon_\theta$ by minimizing the final loss. (v) After $N_{unlearn}$ optimization steps, return the unlearned model.

---

**Algorithm 2:** Pseudo code of SMUD.

---

**Input** : Retain set $D_r$; Unlearning set $D_u$; pre-trained approximator $\epsilon_{\boldsymbol{\theta}}$; $\boldsymbol{\theta}_{\epsilon_\theta}$ denotes parameters of $\epsilon_{\boldsymbol{\theta}}$

**Output:** Unlearned approximator $\epsilon_{\boldsymbol{\theta}}$

**1 for** $n \in \{1, 2, \cdots, N_{\text{unlearn}}\}$ **do**

**2**     Sample $\mathbf{x}_0^r$ from $D_r$

    `// Retain loss `**`Eq.(9)`**` and `**`Eq.(12)`**`;`

**3**     **if** *SMUD for unconditional DMs* **then**

**4**       $l = \left\| \boldsymbol{\epsilon} - \epsilon_{\boldsymbol{\theta}} \left( \sqrt{\bar{\alpha}_t}\mathbf{x}_0^r + \sqrt{1 - \bar{\alpha}_t}\boldsymbol{\epsilon}, t \right) \right\|^2$

**5**     **else if** *SMUD for conditional DMs* **then**

**6**       $l = \left\| \boldsymbol{\epsilon} - \epsilon_{\boldsymbol{\theta}} \left( \sqrt{\bar{\alpha}_t}\mathbf{x}_0^r + \sqrt{1 - \bar{\alpha}_t}\boldsymbol{\epsilon}, t, \mathbf{c} \right) \right\|^2$

**7**     **if** $n \% N_{\text{interval}} = 0$ **then**

**8**       $\epsilon'_{\boldsymbol{\theta}} = \text{copy}(\epsilon_{\boldsymbol{\theta}})$

**9**       Sample $\mathbf{x}_0^u$ from $D_u$

      `// Unlearning loss `**`Eq.(8)`**` and `**`Eq.(11)`**`;`

**10**       **if** *SMUD for unconditional DMs* **then**

        $\mathbf{y} = (\sqrt{\bar{\alpha}_t}\mathbf{x}_0^u + \sqrt{1 - \bar{\alpha}_t}\boldsymbol{\epsilon}, t)$

**11**         $l = l + \left\| \epsilon'_{\boldsymbol{\theta}}(\mathbf{y}) + \gamma\boldsymbol{\epsilon}' - \epsilon_{\boldsymbol{\theta}}(\mathbf{y}) \right\|^2$

**12**       **else if** *SMUD for conditional DMs* **then**

        $\mathbf{y_c} = (\sqrt{\bar{\alpha}_t}\mathbf{x}_0^u + \sqrt{1 - \bar{\alpha}_t}\boldsymbol{\epsilon}, t, \mathbf{c})$

        $\mathbf{y}_\emptyset = (\sqrt{\bar{\alpha}_t}\mathbf{x}_0^u + \sqrt{1 - \bar{\alpha}_t}\boldsymbol{\epsilon}, t, \emptyset)$

**13**         $l = l + \left\| \epsilon'_{\boldsymbol{\theta}}(\mathbf{y_c}) + \gamma\boldsymbol{\epsilon}' - \epsilon_{\boldsymbol{\theta}}(\mathbf{y_c}) \right\|^2 +$

           $\left\| \epsilon'_{\boldsymbol{\theta}}(\mathbf{y}_\emptyset) + \gamma\boldsymbol{\epsilon}' - \epsilon_{\boldsymbol{\theta}}(\mathbf{y}_\emptyset) \right\|^2$

    `// Optimization step;`

**14**     $\boldsymbol{\theta}_{\epsilon_\theta} \leftarrow \boldsymbol{\theta}_{\epsilon_\theta} - \eta\nabla_{\boldsymbol{\theta}_{\epsilon_\theta}} l$

**15 Return** $\epsilon_{\boldsymbol{\theta}}$

---

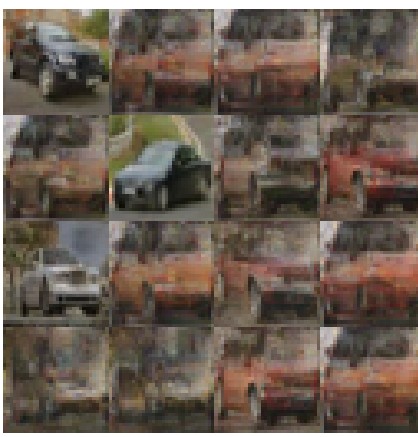 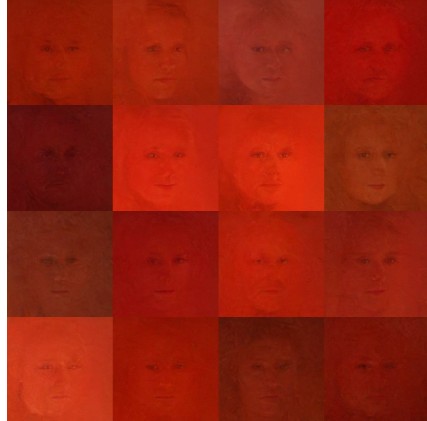

(a) 1.6K unlearning steps (CIFAR10)  (b) 90 unlearning steps (celeb-HQ)

Figure 5: Synthetic images generated by unconditional DDPMs unlearned by SISS.

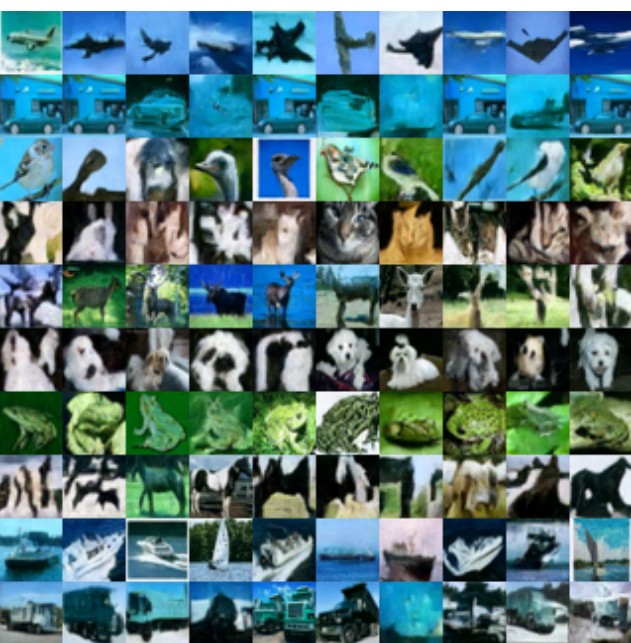

Figure 6: Synthetic images generated by conditional DDPMs unlearned by SISS for 70 steps.

## D    SUPPLEMENTAL RESULTS OF SISS

In this section, we explain why the unlearning process of SISS must be halted earlier. Figure 5a shows the synthetic images generated by the unconditional DDPM after being unlearned for 1.6K optimization steps with SISS on the CIFAR-10 dataset. Figure 5b shows the synthetic images generated by the unconditional DDPM after being unlearned for 90 optimization steps with SISS on the Celeb-HQ dataset. Besides, Fig. 6 shows the synthetic images generated by the conditional DDPM after being unlearned for 70 optimization steps with SISS on the CIFAR-10 dataset. As shown in the above synthetic images, the quality of the synthetic images is too poor, so the unlearning process needs to be stopped.

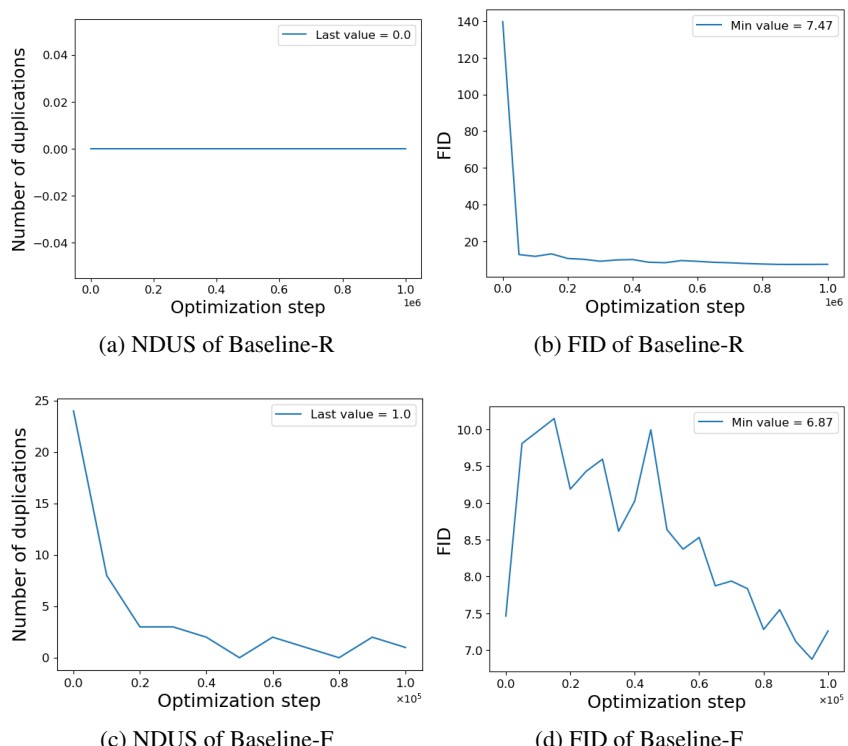

Figure 7: NDUS and FID curves during the unlearning of unconditional DDPM using Baseline-R and Baseline-F.

## E  SUPPLEMENTAL RESULTS FOR UNCONDITIONAL DM

In this section, we demonstrate detailed results when unlearning Automobile. Figure 7 shows the NDUS and FID curves during the unlearning of unconditional DDPM using Baseline-R and Baseline-F. As shown in Fig. 7a, Baseline-R generates zero duplicates of the unlearning set images, retraining the DM from scratch on the retain set. Figure 7b shows that Baseline-R achieves a minimum FID score of 7.47 during 1M optimization steps. On the other hand, Baseline-F fails to unlearn within 100K optimization steps since it still generates a duplicate of an unlearning set image after 100K optimization steps, as shown in Fig. 7c. Figure 7d shows that fine-tuning the pre-trained DDPM on the retain set can decrease the FID score.

Figure 8 shows the NDUS and FID curves during the unlearning of unconditional DDPM using the proposed SMUD. Figure 8a shows the NDUS curves of various $\gamma$ values over the optimization steps. As shown in Fig. 8a, the DDPM successfully unlearns the images from the unlearning set after 3K optimization steps for all $\gamma$ values, and the value of $\gamma$ has a limited influence on the unlearning speed. Figure 8b shows the FID curves over the optimization steps of various $\gamma$. As shown in Fig. 8b, FID scores of the DDPM fluctuate during the unlearning process and the value of $\gamma$ has a limited influence on the amplitude of the fluctuation. The fluctuation in NDUS curves occurs because the DM has not fully unlearned the unlearning set. Consequently, the retain loss, which aids in improving generative performance, leads to increases in NDUS. The fluctuation in FID curves arises because the unlearning loss can reduce generative performance, which is why the retain loss is necessary. To evaluate the existence of fake unlearning, we finetune the unlearned DDPM with the retain loss on the retain set. Figure 9a shows the NDUS curves during the fine-tuning of the DDPMs, which have been unlearned by SMUD for $\{1K, 2K, 3K, 4K, 5K\}$ optimisation steps with $\gamma = 1.0$. The fake unlearning exists when the DDPM is unlearned for 1K optimization steps. After more unlearning optimization steps, the fake unlearning phenomenon disappears, which means a complete unlearning. On the other hand, fine-tuning the unlearned DDPM improves the DDPM's generative performance w.r.t. FID score, as shown in Fig. 9b.

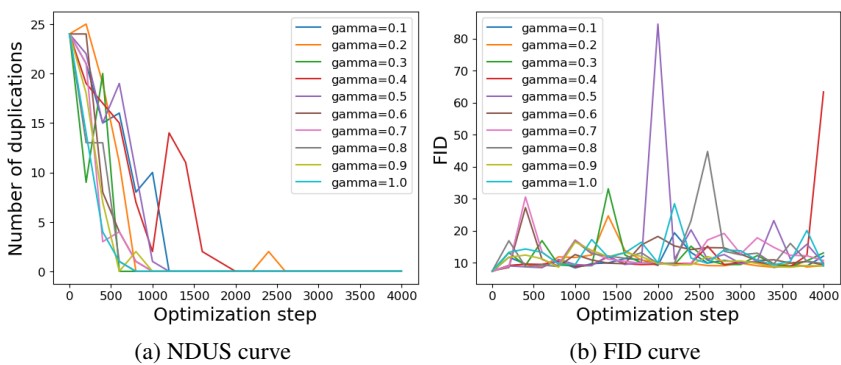

(a) NDUS curve

(b) FID curve

Figure 8: NDUS and FID curves while unlearning the unconditional DDPM using SMUD.

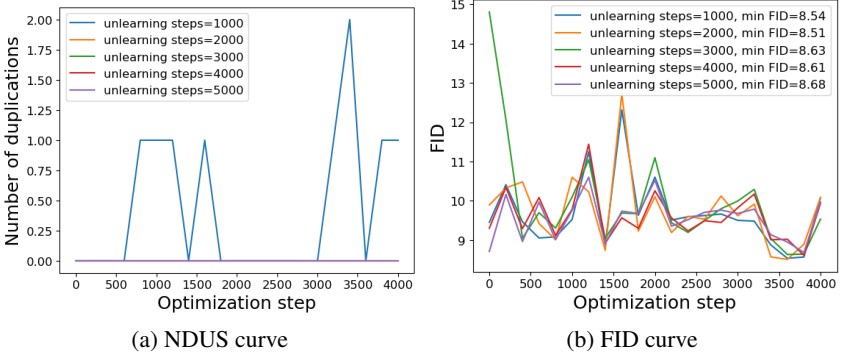

(a) NDUS curve

(b) FID curve

Figure 9: NDUS and FID curves while fine-tuning the unlearned unconditional DDPM after being unlearned for 1K–5K optimization steps using SMUD with $\gamma = 1.0$.

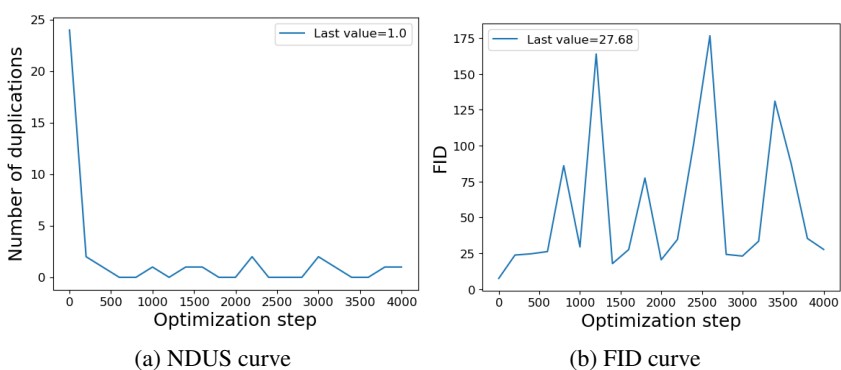

(a) NDUS curve

(b) FID curve

Figure 10: NDUS and FID curves while unlearning the unconditional DDPM using Baseline-GA.

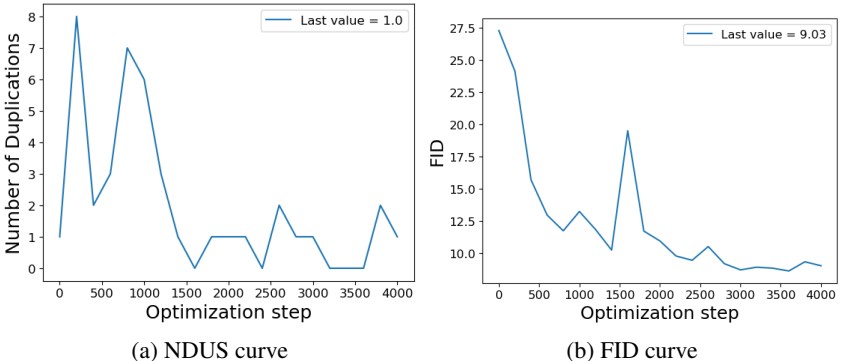

(a) NDUS curve

(b) FID curve

Figure 11: NDUS and FID curves while fine-tuning the unlearned unconditional DDPM after being unlearned for 4K optimization steps using Baseline-GA.

Figure 10 presents the NDUS and FID curves during unlearning unconditional DDPM using Baseline-GA. As shown in Fig. 10a, Baseline-GA fails to unlearn within 4K optimization steps since it still generates a duplicate of an unlearning set image after 4K optimization steps. Furthermore, as shown in Fig. 10b, the fluctuation in FID during unlearning with Baseline-GA is more pronounced than with SMUD, indicating that Baseline-GA degrades the performance of the DDPM to a greater extent. Moreover, as shown in Fig. 11, fine-tuning the DM unlearned by Baseline-GA can increase NDUS.

Figure 12 presents the NDUS and FID curves during the unlearning of the unconditional DDPM using SISS. As shown in Fig. 12b, SISS significantly reduces generative performance. We only plot the FID curve for up to 1400 unlearning steps, as the DM fails to generate any recognizable images beyond this point (refer to Appendix D for details). Furthermore, as shown in Fig. 12a, despite the significant decrease in generative performance, the DM still generates duplicates of the unlearning data even after 1400 unlearning steps. Furthermore, as illustrated in Fig. 13, fine-tuning the DM unlearned by SISS increases NDUS, suggesting that the unlearning ability of SISS is partly due to the decline in generative performance.

## F    SUPPLEMENTAL RESULTS FOR CONDITIONAL DM

Figure 14 shows the NDUS and FID curves during unlearning conditional DDPM using Baseline-R and Baseline-F. As shown in Fig. 14a, Baseline-R generates zero duplicates of the unlearning set images retraining the DM from scratch on the retain set. Figure 14b shows that Baseline-R achieves a minimum FID score of 27.51 during 1M optimization steps. On the other hand, Baseline-F fails to unlearn the unlearning set within 100K optimization steps since it still generates 5 duplicates of unlearning set images after 100K optimization steps, as shown in Fig. 14c. Unlike the unconditional case, fine-tuning the pre-trained DDPM on the retain set cannot decrease the FID score for conditional DDPM, as shown in Fig. 14d. Besides, unlearning the conditional DDPM is more difficult

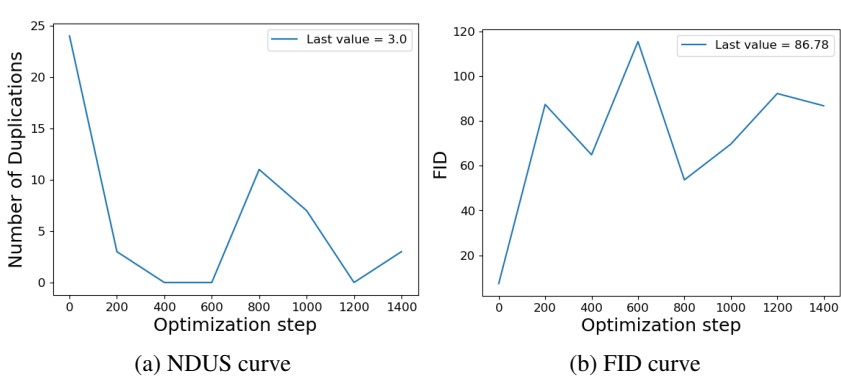

(a) NDUS curve             (b) FID curve

Figure 12: NDUS and FID curves while unlearning the unconditional DDPM using SISS.

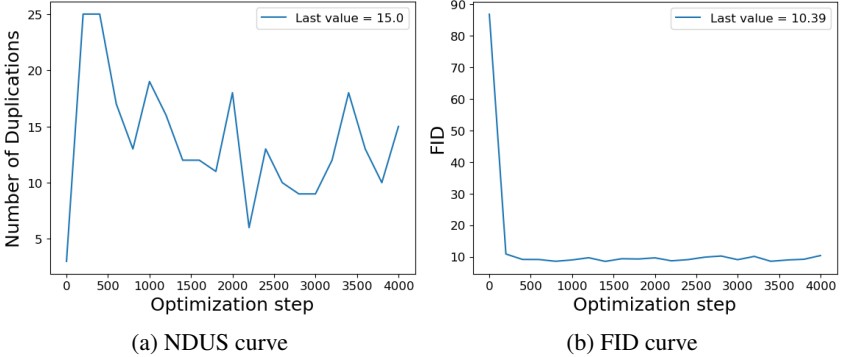

(a) NDUS curve             (b) FID curve

Figure 13: NDUS and FID curves while fine-tuning the unlearned unconditional DDPM after being unlearned for $1.4K$ optimization steps using SISS.

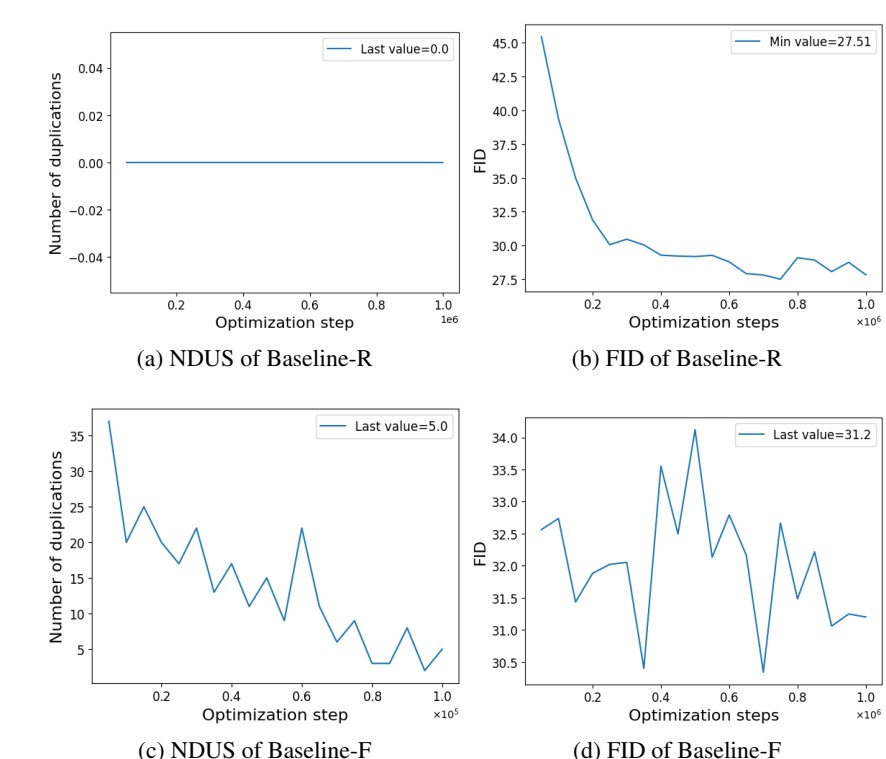

(a) NDUS of Baseline-R

(b) FID of Baseline-R

(c) NDUS of Baseline-F

(d) FID of Baseline-F

Figure 14: NDUS and FID curves during the unlearning of conditional DDPM using Baseline-R and Baseline-F.

than the unconditional DDPM, as evident from the comparison of Fig. 14c and Fig. 7c. Therefore, we perform 10K/20K optimization steps for unlearning in the conditional DDPM instead of 4K steps for the unconditional case.

Figure 15 shows the NDUS and FID curves during the unlearning of conditional DDPM using the proposed SMUD. Figure 15a shows the NDUS curves of various $\gamma$ values over the optimization steps. As shown in Fig. 15a, the DDPM successfully unlearns the unlearning set images after 2K optimization steps for all $\gamma$ values, and the value of $\gamma$ has a limited influence on the unlearning speed. Figure 15b shows the FID curves over the optimization steps of various $\gamma$ values. As shown in Fig. 15b, FID scores of the DDPM fluctuate during the unlearning process and the value of $\gamma$ has a limited influence on the amplitude of the fluctuation. The fluctuation of the FID curves is because the unlearning loss can decrease the generative performance of the DM. To evaluate the existence of fake unlearning, we finetune the unlearned DDPM on the retain set. Figure 16a shows the NDUS curves during fine-tuning the DDPMs, which have been unlearned by SMUD for $\{2K, 4K, 6K, 8K, 10K\}$ optimisation steps with $\gamma = 0.1$. The fake unlearning phenomenon exists when the DDPM is unlearned for less than 8K optimization steps. After unlearning the DM over 8K optimization steps, the fake unlearning phenomenon disappears. On the other hand, fine-tuning after unlearning can help to improve the model's generative performance w.r.t. FID score, as shown in Fig. 16b.

Figure 17 presents the NDUS and FID curves during unlearning conditional DDPM using Baseline-GA. Unlike the unconditional case, NDUS reaches 0 after 500 optimization steps, as shown in Fig. 17a. Then, we finetune the unlearned DDPM for 10K optimization steps, by minimizing the retain loss on the retain set. As shown in Fig. 18, NDUS increases significantly during the fine-tuning, which indicates that Baseline-GA did not achieve complete unlearning.

Figure 19 presents the NDUS and FID curves during unlearning conditional DDPM using SISS. Similar to the unconditional case, SISS significantly decrease the generative performance, as shown in Fig. 19b. We only plot the FID curve for up to 60 unlearning steps, as the DM fails to generate

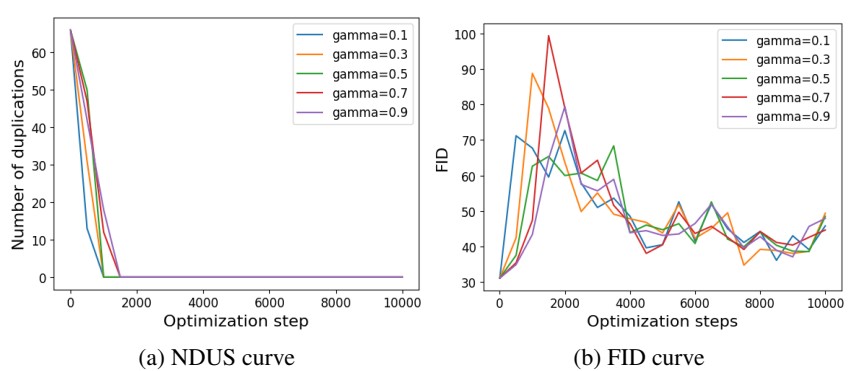

(a) NDUS curve        (b) FID curve

Figure 15: NDUS and FID curves while unlearning the conditional DDPM using the proposed SMUD.

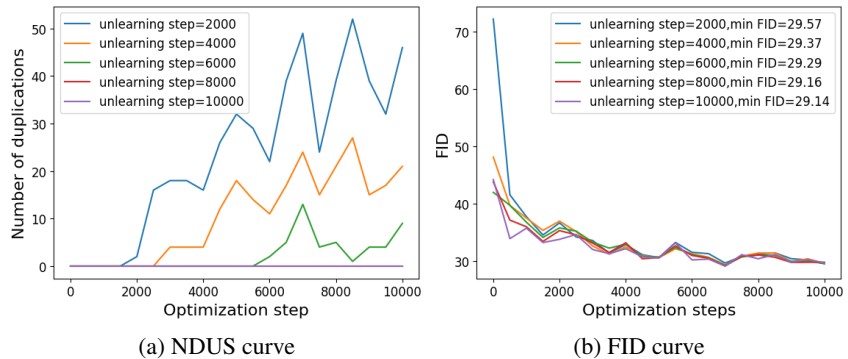

(a) NDUS curve        (b) FID curve

Figure 16: NDUS and FID curves during fine-tuning the unlearned conditional DDPM after being unlearned for 2K–10K optimization steps using SMUD with $\gamma = 0.1$.

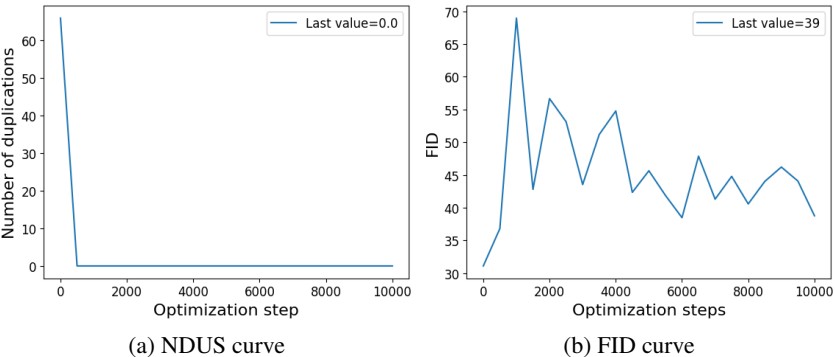

(a) NDUS curve        (b) FID curve

Figure 17: NDUS and FID curves while unlearning the conditional DDPM using Baseline-GA.

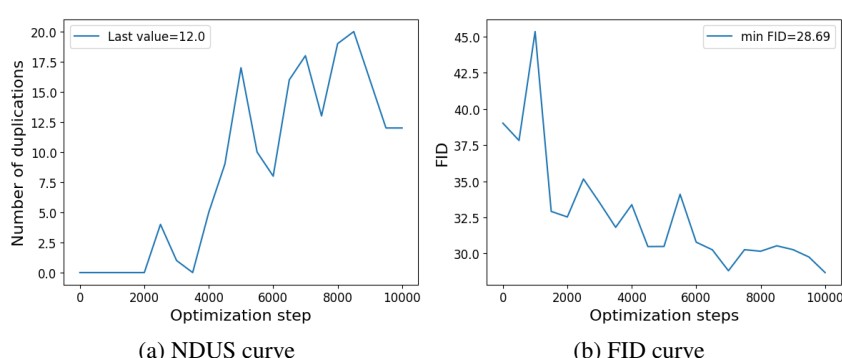

(a) NDUS curve

(b) FID curve

Figure 18: NDUS and FID curves while fine-tuning the unlearned conditional DDPM after being unlearned for 10K optimization steps using Baseline-GA.

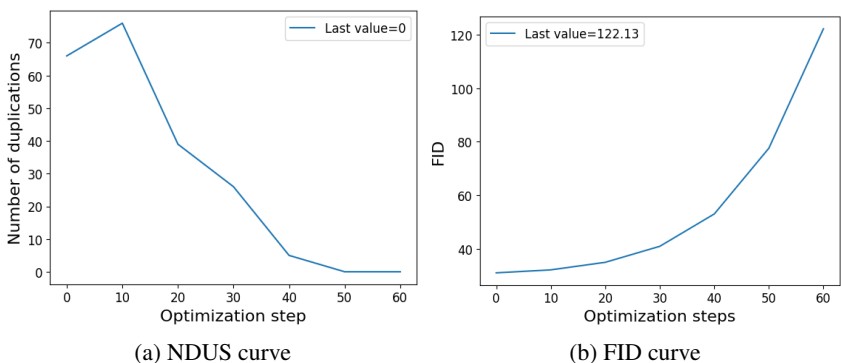

(a) NDUS curve

(b) FID curve

Figure 19: NDUS and FID curves while unlearning the conditional DDPM using SISS.

any recognizable images beyond this point (refer to Appendix D for details). Moreover, as shown in Fig. 20, fine-tuning the DM unlearned by SISS increases NDUS, indicating that SISS did not achieve complete unlearning.

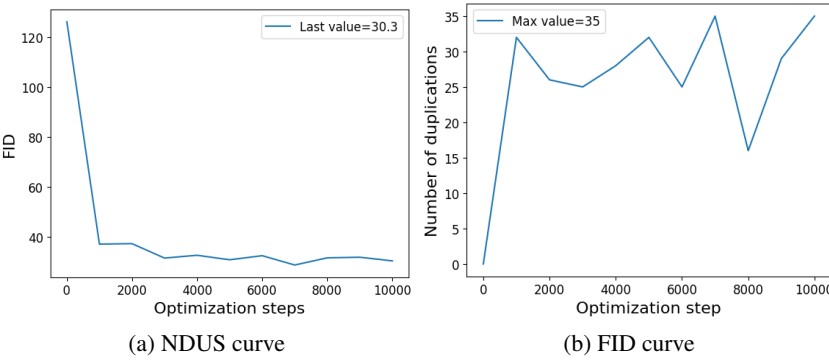

(a) NDUS curve

(b) FID curve

Figure 20: NDUS and FID curves while fine-tuning the unlearned conditional DDPM after being unlearned for 60 optimization steps using SISS.

## G  SUPPLEMENTAL RESULTS OF SMUD ON CELEBA-HQ

Following the qualitative evaluation framework introduced in Section 3.2, we reconstructed the unlearning set images from partially noised images after 400 forward steps. Figure 21 shows the unlearning set images, synthetic images reconstructed by the pre-trained DDPM and Baseline-R. The synthetic images reconstructed by the pre-trained DDPM closely resemble the corresponding unlearning set images, comparing Fig. 21a and Fig. 21b. Although Baseline-R is not trained on the unlearning set images, it can still reconstruct similar images because the partially noised images retain some information about the original images, as shown in Fig. 21c. However, the differences between Fig. 21c, and Fig. 21a are more pronounced compared to those between Fig. 21b and Fig. 21a if we zoom in.

Figure 22a and 22b show synthetic images reconstructed by the DDPM unlearned by Baseline-GA and SMUD, respectively, after 100K optimization steps. Consistent with Fig. 4, compared with the original unlearning images, the images reconstructed by SMUD show greater differences than Baseline-GA, highlighting its superiority over Baseline-GA.

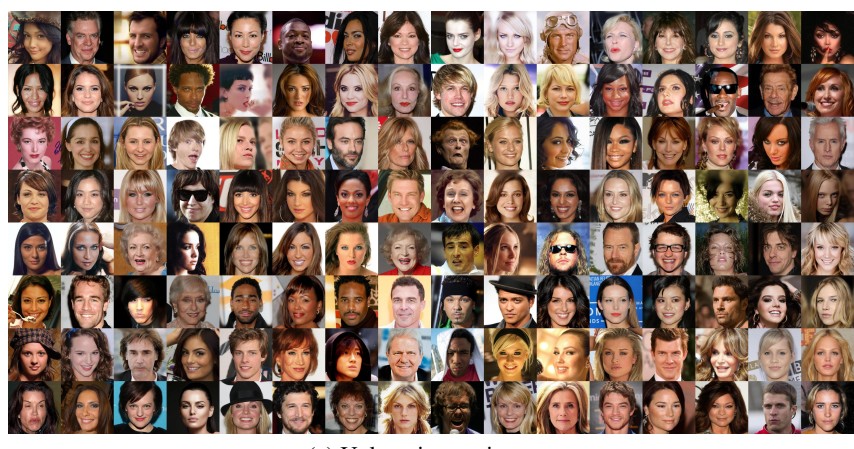

(a) Unlearning set images

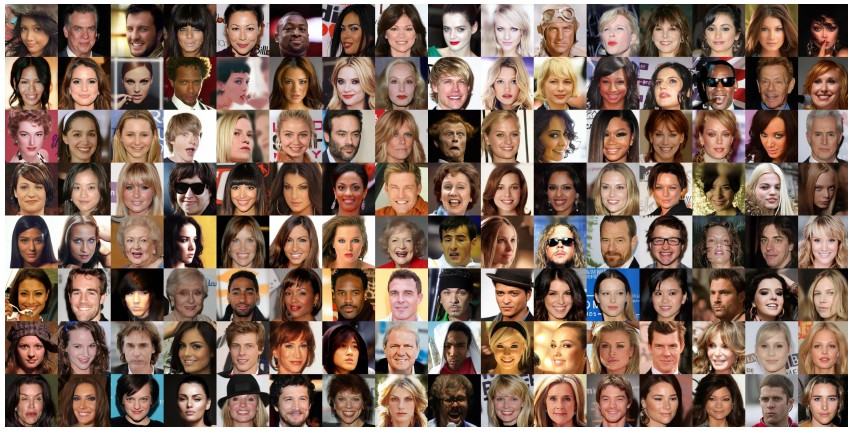

(b) Reconstructed by the pre-trained DDPM

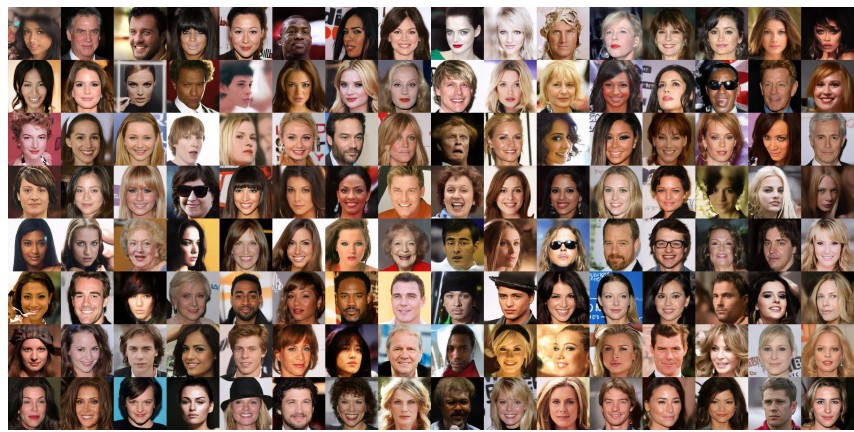

(c) Reconstructed by Baseline-R (trained on the retain set)

Figure 21: The unlearning set images and synthetic images reconstructed by the pre-trained DDPM and Baseline-R.

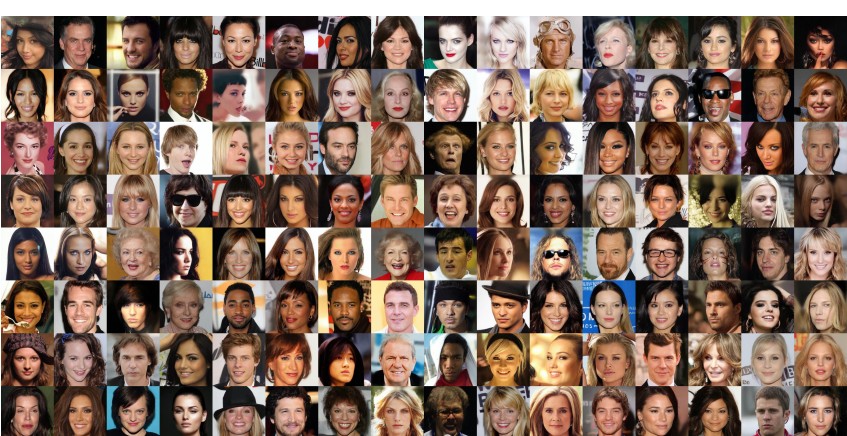

(a) Reconstructed by Baseline-GA

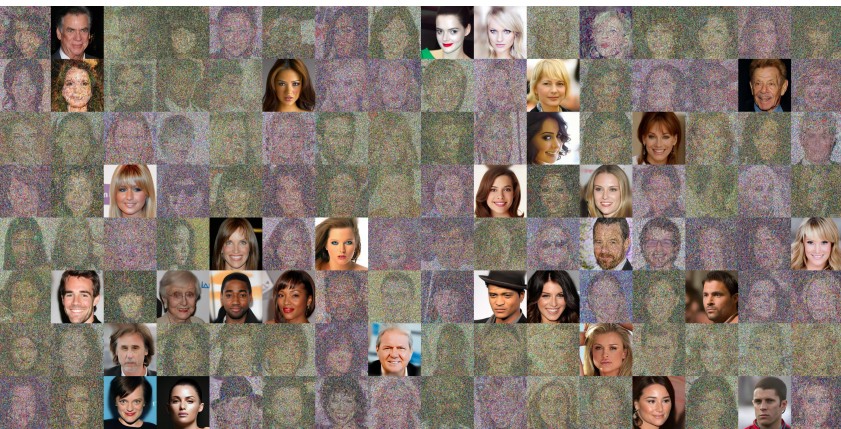

(b) Reconstructed by SMUD

Figure 22: Synthetic images reconstructed by the DDPMs unlearned with Baseline-GA and SMUD.

