# OpenReview forum: "UNLEARNING TRAINING DATA FROM DIFFUSION MODELS"
_ICLR.cc/2026/Conference — ICLR 2026 Conference Withdrawn Submission_

### Official Review · Reviewer_9JSE · 2025-10-29

**Soundness:** 3
**Presentation:** 3
**Contribution:** 3
**Rating:** 6
**Confidence:** 3

**Summary:**

This paper investigates sample-level machine unlearning (SLMU) in diffusion models, a setting motivated by privacy and regulatory concerns (e.g., GDPR) around the removal of individual training samples from generative models. Drawing attention to the shortcomings of class-level unlearning and the limitations of prior work (notably SISS), the authors propose: (1) a formal objective and new benchmark for SLMU in DMs, focused on the Number of Duplicates of the Unlearning Set (NDUS) to detect memorization traces post-unlearning and after additional fine-tuning; (2) the identification of a “fake unlearning” phenomenon, whereby models rememorize samples after finetuning; (3) a new method, SMUD, for SLMU on unconditional and conditional DMs, which perturbs the generative path to erase targeted sample memorization while maintaining fidelity on the retain set. Extensive experiments are conducted on CIFAR-10 and CelebA-HQ, with detailed quantitative and qualitative evaluations, baseline comparisons, and ablation analyses.

**Strengths:**

1. The introduction of "fake unlearning", where unlearned models memorize removed samples after restoration of performance via additional finetuning, is a sharp and valuable empirical observation, immediately useful to the research community.

2. They propose an actionable and quantifiable evaluation protocol (Section 3.2 and Appendix B), centering NDUS, which better detects both lasting memorization and fake unlearning, a subtle failure mode that might go unnoticed otherwise.

3. The SMUD approach is mathematically transparent, principled, and integrates seamlessly into standard DM workflows, as shown by the stepwise pseudo-code in Appendix C and clear equations in Section 4 (unlearning/retain loss, unlearning path modification).

4. Experiments are comprehensive in scope: ablation over noise magnitudes (γ), side-by-side evaluations of unconditional and conditional DMs, and baseline comparisons across multiple CIFAR-10 classes and CelebA-HQ. The key results tables (Table 1 and Table 2) robustly support the claims, showing that SMUD most effectively reduces NDUS and avoids fake unlearning, while maintaining competitive FID scores.

**Weaknesses:**

1. Table 2 (conditional DMs) reveals that even the best FID scores are substantially higher (worse) than unconditional DMs, with significant variance by class. There is little discussion explaining this gap or the tradeoffs for conditional generation.

2. The “fake unlearning” phenomenon is important, but under-explored. For instance, why do some baselines exhibit it more strongly? Can NDUS distinguish between partial vs. full rememorization during fine-tuning?

3. The realism of unlearning sets composed solely of the most-duplicated images (the “easy cases” for SLMU) is questionable; practical privacy queries may not always concern the most-memorized samples.

**Questions:**

1. Metric Sensitivity and Generalizability: How does the NDUS metric perform for unlearning sets drawn from more typical or "average" samples (not just the most-memorized)? Does it generalize to lower-memorization regimes or larger, more diverse real-world datasets where duplicates may not be as frequent?

2. Alternative Detection of Memorization: Have the authors considered alternative measures (perceptual similarity, feature embedding analysis) for monitoring memorization, especially for small modifications or partial reconstructions? How robust is NDUS to adversarial attempts to evade detection?

3. Unlearning in "Hard" vs. "Easy" Cases: Is the proposed method or evaluation protocol effective when the unlearning set is composed of challenging, low-memorization, or out-of-distribution samples? Does the approach degrade in such cases?

---

### Official Review · Reviewer_XVXD · 2025-10-29

**Soundness:** 2
**Presentation:** 3
**Contribution:** 2
**Rating:** 2
**Confidence:** 3

**Summary:**

This paper studies sample‐level machine unlearning (SLMU) in diffusion models (DMs), aiming to selectively remove the influence of specific training samples. The authors first critique previous work (notably SISS) for inadequate evaluation metrics and introduce a new quantitative evaluation framework leveraging diffusion model memorization to detect “fake unlearning.” They propose SMUD, a new unlearning algorithm that modifies the reverse diffusion process by injecting Gaussian noise into the denoising steps, thereby altering the generative trajectory of memorized samples.

**Strengths:**

- The paper identifies and articulates a genuine problem in evaluating unlearning for generative models, contributing a clearer definition and metric (NDUS).

- The proposed SMUD method is novel in its design—injecting controlled stochastic noise into diffusion trajectories is an original and intuitive mechanism.

- The experiments include both unconditional and conditional DMs, showing consistent empirical trends.

- The paper is technically detailed and provides mathematical formulations for reproducibility.

**Weaknesses:**

- Recent literature (" The Illusion of Unlearning: The Unstable Nature of Machine Unlearning in Text-to-Image Diffusion Models") already exposes critical vulnerabilities in diffusion model unlearning—e.g., methods showing that “unlearned” concepts can resurface under benign fine-tuning ("concept resurgence"). The paper’s contribution may be incremental and not sufficiently address the larger robustness gap in unlearning for DMs.

- While the proposed NDUS metric is creative, its dependence on memorization detection from synthetic data is questionable. It assumes duplications correspond directly to memorization, but lacks validation against true data‐removal guarantees. How robust is NDUS to noise or latent similarity thresholds?

- The paper lacks a formal proof or theoretical analysis linking the proposed loss functions (Eq. 8 and 11) to successful unlearning beyond empirical correlation. Can the authors provide convergence or generalization arguments?

- Experiments are limited to small datasets (CIFAR-10, CelebA-HQ). There is no evaluation on large text‐to‐image or multimodal DMs (e.g., Stable Diffusion), where scalability and privacy risks are more relevant.

- The phenomenon is defined qualitatively, and the detection relies solely on FID and NDUS changes post fine‐tuning. This approach could misinterpret partial forgetting as complete unlearning. Can the authors propose quantitative robustness checks?

**Questions:**

Refer to the Weakness above.

---

### Official Review · Reviewer_2dq3 · 2025-10-31

**Soundness:** 2
**Presentation:** 3
**Contribution:** 2
**Rating:** 4
**Confidence:** 4

**Summary:**

In this paper, the authors focus on the problem of Sample-Level Machine Unlearning (SLMU) for diffusion models, which targets the removal of individual training data. Observing the fake unlearning phenomenon, where unlearned samples can be reproduced after finetuning on the retain set, they propose a new evaluation framework to adequately assess the SLMU performance. Additionally, the authors propose a finetuning approach to address the fake unlearning problem, which is supported by the experimental results using the proposed evaluation framework.

**Strengths:**

1. The proposed sample-level machine unlearning method is simple and effective. The idea of modifying the generation path of DMs is interesting.
2. The experimental results demonstrate that the proposed method is indeed more robust than the baselines.

**Weaknesses:**

1. Although the paper points out the "fake unlearning" phenomenon, it does not fully explain why fine-tuning on the retain set can restore the unlearned samples. Considering the special setting of a small training dataset, unlearning methods using unlearning samples from the original training dataset might not be sufficient. A simple experiment would be to use synthetic variations of the unlearning samples for unlearning.
2. The proposed unlearning loss is mainly heuristic without detailed analysis and theoretical guarantees. Despite the loose connection to Equation 7, minimizing the actual unlearning loss of SMUD defined in Equation 8 itself does not explain why it can effectively alter the generation path of the unlearning images. In Equation 8, $x_0, \epsilon,$ and $\epsilon'$ are drawn independently, so the optimal model weights $\theta^\*$ for the objective will be the posterior expectation, *i.e.,* $\epsilon_{\theta^*}(x_t, t)=\mathbb{E}[\epsilon'(x_t,t)+\gamma\epsilon'|x_t]=\epsilon'(x_t,t)$. This means that minimizing this unlearning loss does not guarantee that the (optimal) unlearned model can alter the generation path of unlearning samples.
3. The FID comparison in Figure 4c  is counter-intuitive. The "gold standard" Baseline-R (retrained from scratch) shows a *worse* FID than the SMUD-unlearned model. This implies either the 800K-step training protocol for Baseline-R was not converged, or the extra 100K steps for SMUD  provided an unfair fine-tuning advantage, making the comparison misleading.

**Questions:**

1. Could the authors please explain the FID result in Figure 4c? Why does the SMUD-unlearned model achieve a better FID than both the pre-trained model and the "gold standard" Baseline-R model? If this is due to the 800K-step models not being fully converged, doesn't this invalidate the comparison, as SMUD received an extra 100K-step fine-tuning?

2. The reconstructions in Figure 4b suggest SMUD learns not to properly denoise samples from $D_u$. However, it is hard to tell if SMUD affects the model's generalizability. For example, the model trained on SMUD loss may perform poorly on unseen images that are similar to the unlearned samples or sampled from the same data distribution as these unlearned samples.

3. In the SMUD algorithm (Eq. 8), the target $\epsilon'\_{\theta}$ is a copy of $\epsilon\_\theta$ and is updated every $N_{\text{interval}}$ steps. What is the motivation for this "moving target"? Would the method be more or less effective if $\epsilon'_{\theta}$ was permanently frozen to the original, pre-trained model $\theta_p$?

4. It seems that the SMUD algorithm could also be used for class-level unlearning. Have the authors tested it? If so, how does the performance compare with other methods.

---

### Official Review · Reviewer_pQSL · 2025-11-01

**Soundness:** 3
**Presentation:** 3
**Contribution:** 3
**Rating:** 6
**Confidence:** 4

**Summary:**

This paper addresses the problem of sample-level machine unlearning (SLMU) in diffusion models, which is critical for privacy and copyright protection. The authors define the objectives of SLMU for diffusion models and propose a quantitative evaluation framework based on the Number of Duplicates of the Unlearning Set (NDUS) metric, identifying the "fake unlearning" phenomenon. They introduce a novel method, SMUD, which modifies the generative path of target images by adding noise to the reverse process. Experiments demonstrate that SMUD achieves effective unlearning without fake unlearning while preserving generative performance.

**Strengths:**

1. The setting of the SLMU task is interesting and very practical.

2. The proposed method (SMUD) is simple but effective.

**Weaknesses:**

+ The paper tackles the Selective Machine Unlearning (SLMU) task, but it's unclear how one determines which specific samples should be selected for unlearning in a practical scenario. Could the authors elaborate on the intended use case? For example, does this assume a user can identify specific images from their data that must be forgotten?

+ The experiments are conducted on relatively small-scale datasets like CIFAR-10. How well does the proposed method scale to larger, more complex datasets (e.g., ImageNet) and to larger diffusion model architectures?

+ Is the SMUD method effective for modern, large-scale text-to-image diffusion models such as Stable Diffusion or FLUX? Demonstrating its performance on these models would significantly broaden the impact of the work.

+ The number of selected images is currently set to 16 (Line 357). How sensitive are the results to this hyperparameter? An ablation study on the effect of this number would be very informative.

+ To better understand the results, it would be useful to see the performance (e.g., FID score) of a model trained from scratch without the unlearning set. This would provide a clear upper bound for the ideal "exact unlearning" outcome.

**Questions:**

See Weaknesses.

---

### Note · Authors · 2025-11-27

I have read and agree with the venue's withdrawal policy on behalf of myself and my co-authors.